# Overview of Dual-Acting Drug Methotrexate in Different Neurological Diseases, Autoimmune Pathologies and Cancers

**DOI:** 10.3390/ijms21103483

**Published:** 2020-05-14

**Authors:** Przemysław Koźmiński, Paweł Krzysztof Halik, Raphael Chesori, Ewa Gniazdowska

**Affiliations:** Centre of Radiochemistry and Nuclear Chemistry, Institute of Nuclear Chemistry and Technology, Dorodna 16, 03-195 Warsaw, Poland; P.Halik@ichtj.waw.pl (P.K.H.); r.chesori@ichtj.waw.pl (R.C.); e.gniazdowska@ichtj.waw.pl (E.G.)

**Keywords:** methotrexate, myasthenia gravis, Alzheimer’s disease, immunosuppressive drug, disease modifying anti-rheumatic drug, anticancer drug

## Abstract

Methotrexate, a structural analogue of folic acid, is one of the most effective and extensively used drugs for treating many kinds of cancer or severe and resistant forms of autoimmune diseases. In this paper, we take an overview of the present state of knowledge with regards to complex mechanisms of methotrexate action and its applications as immunosuppressive drug or chemotherapeutic agent in oncological combination therapy. In addition, the issue of the potential benefits of methotrexate in the development of neurological disorders in Alzheimer’s disease or myasthenia gravis will be discussed.

## 1. Introduction

Methotrexate (MTX, amethopterin, Figure 1) is a synthetic organic compound, which belongs to an anti-folate therapeutic group and to class III of the Biopharmaceutical Classification System. It was created in the late 1940s as a less toxic derivative of aminopterin (Figure 1), then a folic acid (FA) antagonist used to treat children with acute leukaemia [1]. A few years later, it was noted that a low dose of aminopterin (1–2 mg/day) causes significant improvement in patients with rheumatoid arthritis (RA) [2] and in patients with psoriasis [3]. However, it was MTX that was introduced to clinical application in RA since the mid-1980s [4,5]. Currently, MTX is commonly applied in combination with other drugs for the treatment of many neoplasms (acute lymphoblastic leukaemia, acute myeloid leukaemia, meningeal leukaemia and lymphoma, osteosarcomas, non-Hodgkin’s lymphoma, also breast, bladder and number of other cancers) [6,7,8,9,10,11,12,13], severe and resistant forms of autoimmune diseases (rheumatoid arthritis, psoriasis, myasthenia gravis, Crohn’s disease, multiple sclerosis, polyarticular juvenile idiopathic arthritis) [4,5,6,7,14,15,16,17,18,19], or even an ectopic pregnancy [20]. 

Particularly, the introduction of low dose methotrexate (LDMTX) therapy of RA and psoriasis with dose of 7.5–25 mg/week versus high dose methotrexate (HDMTX) therapy of 1–5 g/week in cancer therapy became great breakthrough [15]. This approach was found to be relatively safe (especially in case of serious interactions with other drugs) and significantly reduced the occurrence of relevant adverse effects [6,7], what highly improved patient tolerance and therapy compliance. Since then, perception of MTX in the clinical environment has changed; moreover, this drug became the gold standard for the treatment of RA [21,22], demonstrating greater efficacy and safety than other synthetic disease-modifying anti-rheumatoid drugs (DMARDs), while biological drugs became only a complement to MTX application. The clinical success of MTX has prompted a further search for new multi-functional dihydrofolate reductase (DHFR) antagonists [23,24,25]. Over the past two decades, many natural and synthetic DHFR antagonists have been discovered and have already been registered mainly for oncological indications; however, MTX is still widely used in the treatment of various diseases and has not been allowed to become a thing of the past.

This review will present MTX in terms of its broad clinical use, application in the therapy of autoimmune diseases, including central nervous system disorders like myasthenia gravis (MG) or Alzheimer’s disease (AD) and application in oncological combination therapy with other drugs.

## 2. Methotrexate—Mechanisms of Drug Action

MTX is an anti-metabolite (anti-vitamin) of folic acid (FA, vitamin B9), which acts as anticancer agent and immunosuppressant [26,27]. MTX indirectly inhibits cell division through the blockage of folate-related enzymes, mainly DHFR, that catalyses the conversion of dihydrofolate to tetrahydrofolate (THF). THF serves as a significant coenzyme in several transmethylation reactions in pyrimidine and purine nucleotide synthesis pathways, essential in synthesis, repair or replication of DNA strands [28,29]. Actually, the methyl-THF acts as proximal methyl donor in numerous methylation reactions of DNAs, RNAs, proteins, phospholipids and amino acids syntheses. Inhibition of intracellular THF production by MTX results in disruption of cell proliferation and its metabolic imbalance.

MTX crosses the biological barriers very poorly, being highly ionized and generally hydrophilic. Bioavailability and biodistribution of the drug are determined by an active transport system [30,31]. Intestinal tissue adsorption of MTX occurs by the proton-coupled folate transporters (PCFTs), which are a solute carrier transporter, while a cellular drug penetration is followed mainly by the reduced folate carrier 1 (RFC1), an APT-binding cassette transporter. To a small extent, MTX also uses receptor-mediated endocytosis via folate receptors (FRs), the glycosyl-phosphatidyl-inositol (GPI)-anchored membrane proteins that may internalize bound folates and folate conjugates [32,33]. Intracellularly, MTX is metabolized by folylpolyglutamyl synthase (FPGS) to a polyglutamate derivatives (MTX_Glu_), that show significantly increased cell residence time and bioactivity in comparison to initial MTX form (Figure 2) [34,35,36]. This is a key pharmacokinetic step that determines the attributed effect of this drug, defining MTX as a representative type 1 prodrug, that undergoes bioactivation inside the cell [37]. Polyglutamated MTX is a superior anti-folate agent than MTX, capable of highly potent DHFR inhibition. Moreover, it also induces inhibition of other enzymes like thymidylate synthase (TYMS) [38], 5-aminoimidazole-4-carboxamide ribonucleotide transformylase (AICART) [39,40] and amido-phosphoribosyltransferase [41,42] participating in de novo biosynthesis of purine and pyrimidine nucleotides. Consequently, it is MTX_Glu_ that deprives a cell of precursors for the synthesis of DNA and RNA necessary for cell proliferation (Figure 2), leading to DNA synthesis disturbances and subsequent cell apoptosis [43]. It should not be surprising that MTX activity is most visible in actively dividing cells, mainly in the S phase of the cell cycle, and in fact, that highly proliferating cancer cells are the most susceptible to the cytotoxic effect of this drug, indicating that antagonism of folate is related with the anti-tumour activity of MTX.

At the same time, the continuous action of MTX polyglutamates in cellular biochemistry results in intracellular accumulation of 5-aminoimidazole-4-carboxamide ribonucleotide (AICAR) by AICART inhibition. AICAR has an ability to diminish activity of adenosine deaminase (ADA) and adenosine monophosphate (AMP) deaminase (AMPDA) as well (Figure 2) [39]. An MTX-mediated excess of AICAR promotes the AMP and adenosine increase and subsequent release of these adenine derivatives outside the cell. Extracellularly, AMP follows easily dephosphorylation to adenosine via ecto-5′-nucleotidase, also known as the CD73 enzyme [44]. Adenosine, a significant signalling agent that modulates diverse physiological functions, acts as major mediator of anti-inflammatory action associated with MTX. In the local extracellular space, adenosine might interact with its specific receptors (A_1_, A_2A_, A_2B_ or A_3_) present on the surface of the origin tissue, but also on immune system cells (Figure 2). Adenosine receptors belong to the seven-transmembrane receptor family, mediating signals into the cell by coupled specific G proteins depending on the receptor subtype. In RA patients, the occurrence of adenosine-specific receptors on immune and synovial cells is elevated, especially A_2A_ and A_3_ that highly mediate adenosine regulation of immune response and inflammation [45,46,47,48]. The adenosine action on A_2A_ and A_2B_ stimulates intracellular production of cyclic adenosine monophosphate (cAMP), a significant cellular second messenger, while stimulation of A_3_ induces phospholipases C and D or inhibits adenylate cyclase producing cAMP. As a result, adenosine in activated neutrophils and macrophages reduces their ability of adhesion on endothelium and phagocytosis as well as generation of superoxide anions and further reactive oxygen species [49,50]. Moreover, leukocytes decrease the production of tumour necrosis factor α (TNF-α) and interleukin (IL)-12—the mediators of inflammation—and promotes anti-inflammatory IL-4 and IL-10. Additionally, it is observed that an inhibition of IL-1 action and suppression of TNF-α excretion in T cells and macrophages inhibit the prostaglandins and leucotrienes syntheses [50,51,52,53,54,55,56]. At the same time, adenosine disrupts the cytokine-mediated inflammation in endothelial cells by decrease in production of IL-6 and IL-8 and hindering leucocytes’ adhesion [57]. 

Thus, an adenosine-mediated effect seems to be a key mechanism in MTX anti-inflammatory action. Nevertheless, the MTX_Glu_ also has some other impact on immune chemotaxis and reduction of the occurrence of inflammation mediators. Sustained decrease of THF-mediated methylation downregulates an accumulation of spermine and spermidine polyamines in the extracellular fluids and lymphocytes of patients with RA [58,59]. These polyamines are essential cellular growth factors among others in lymphocytes. MTX action inhibits T cell proliferation and synthesis of immunoglobulins or rheumatoid factor in RA patients [60,61], followed by a local reduction in lymphocytes interferon (IFN)-γ and IL-2 production, decrease of inflammation, convergent to adenosine receptor-mediated effects.

## 3. Methotrexate—Applications in Medicine

All of the above indicates that MTX, as a drug, has a pleiotropic mechanism of action. Potent anti-metabolic and anti-proliferating action is related with the therapeutic application of MTX as an anticancer agent. The action leading to a local increase of the adenosine concentration at the site of inflammation and also the specific immunosuppressive effect of MTX play an important role in the treatment of severe inflammation and autoimmune diseases (arthritis, psoriasis, myasthenia gravis). Moreover, MTX may serve as vector leading the MTX-attached agents to specific molecular targets like RFC or, less likely FRs [32,33,62,63], to be overexpressed on various cancer cells. While FA plays the role of a Trojan horse for molecular selective delivery of the FA-attached agent to FR-expressing cancer cells, MTX is considered the cornerstone of anti-folate therapy in the treatment of several malignancies.

### 3.1. Methotrexate—Anti-Inflammatory and Immunosuppressive Drug

Since the 1950s, MTX still remains a great interest as an anti-inflammatory and immunosuppressive drug. Its effectiveness in low dose therapy is so irreplaceable that it has become a basic drug of the disease-modifying anti-rheumatic drug (DMARD) category. Nowadays, DMARDs, like MTX, sulfasalazine, azathioprine (AZT) together with biological drugs (e.g., tocilizumab, infliximab) are used to relieve pain and stop disease progression. MTX is extensively used in rheumatoid arthritis, juvenile idiopathic arthritis, psoriatic arthritis, and other inflammatory disorders like myasthenia gravis (MG), Crohn’s disease, multiple sclerosis, sarcoidosis [64,65,66,67,68,69,70,71,72].

#### 3.1.1. Methotrexate-Based Therapy of Myasthenia Gravis 

MG, meaning “serious muscle weakness”, is a rare autoimmune-neurological disorder more commonly occurring in females than in males. It is manifested by weakness and rapid fatigue of pathologically altered skeletal muscles under voluntary control of the somatic nervous system [73,74,75]. MG usually starts in the extraocular muscles, especially those responsible for lifting of the eyelid. Often, only one eye is affected for a long time. Weakness quickly applies to facial muscles, palate muscles and extremities. In severe disease form, respiratory muscle involvement leads to breathlessness and life-threating breathing disorders [76,77].

Most patients with MG have abnormally elevated levels of antibodies against the acetylcholine receptor, muscle tyrosine kinase or lipoprotein-related protein 4 [78,79,80]. That is why treatments of MG are based on acetylcholinesterase inhibitors pyridostigmine and ambenonium to strengthen cholinergic neurotransmission. They are often combined with immunosuppressive drugs that reduce the overactivity of the immune system in production of antibodies against postsynaptic elements of neuromuscular junction. These drugs include glucocorticosteroids (GCS), AZT, cyclophosphamide and MTX [81], where GCS are the first-choice drugs applied for MG. In case of the absence of a response to the treatment, AZT may be applied or MTX as an alternative in AZT intolerance [82,83,84,85]. 

The study described by Hartman et al. gave information that LDMTX applied in patients suffering from MG was quite effective, safe and well tolerated [86]. A similar point of view is presented by Gold et al., where patients with MG received an MTX dosage of 7.5 to 15 mg orally per week [82,87]. MTX seems to have a beneficial effect in slowing progression of MG without any serious side effects. Abdou et al. showed some improvement in 87% of patients who received LDMTX intramuscular weekly for up to 20 months [88]. In another 24-month, single-blinded trial, authors compared LDMTX together with prednisone versus AZT combined with prednisone. They obtained comparable efficacy and tolerability results in both cases. Based on experimental data, the authors suggest that MTX may be a useful drug for MG treatment [89]. Karaahmet et al. described a significant improvement in the muscle weakness and fatigue of patients with MG after treatment with LDMTX over a year. The authors believe that MTX is an effective drug in the treatment of MG. However, they are aware that detailed research is needed to determine conclusive role of MTX in relation to MG [17]. A randomized, double-blind, placebo-controlled trial of MTX for patients with MG was described by Pasnoor et al. [90]. In this study, patients were treated with LDMTX and prednisone or placebo prednisone tablets. The obtained results showed no benefit of MTX in MG over 12 months of combination treatment. Based on the literature, MTX is a third-line agent for the treatment of MG.

#### 3.1.2. Methotrexate-Based Therapy of Rheumatoid Arthritis 

RA is an immune-mediated chronic disorder of connective tissue system manifesting in pain, stiffness and swelling of the joints, mainly of the hands and feet, but inflammation affects often other joints. No treatment leads to joint damage and severe disability, as well as damage to many organs and even premature death [91,92]. The aetiology of this disease is still not fully understood but involves a complex interplay of environmental and genetic factors. RA is a chronic disease originated by a specific antigen and caused by overreactive memory T cells, which play a key role in the non-specific inflammatory process. Self-activation of the memory T cells causes constant amplification of the disease. These lymphocytes release cytokines as IFN-γ and IL-2, which further activate monocytes and macrophages releasing subsequent IL-1, TNF-α and growth factors [93,94,95,96,97,98].

There are three main groups of drugs used for the management of RA: non-steroidal anti-inflammatory drugs (NSAIDs), corticosteroids and disease-modifying anti-rheumatic drugs. The first two groups aim only to reduce symptoms; however, modern therapy regimens are set to achieve disease remission introducing DMARDs early in treatment. Initiating treatment with DMARDs as soon as possible after diagnosis produces significant clinical and functional benefit for the patient.

MTX is commonly used for the treatment of patients with RA and in several other forms of inflammatory arthritis and autoimmune disease. According to many placebo-controlled trials of MTX versus other approved DMARD’s, MTX was established as the standard of care and first-line therapy drug for RA therapy. Furst et al. [99] confirmed that MTX with a dose of 12.5–20 mg/week had a significantly greater effect than the placebo in RA patients. A similar observation was described by Weinblatt et al. [100]. To increase the therapeutic effect, MTX is also used combined with other drugs like sulfasalazine, hydroxychloroquine and especially biological DMARDs. Based on literature, a combination of MTX with anti-TNF therapy was considerably better than monotherapy with MTX [101,102].

#### 3.1.3. Methotrexate-Based Therapy of Alzheimer’s Disease

Alzheimer’s disease (AD), diagnosed by Alois Alzheimer in 1907, is a disorder of the central nervous system causing general dementia. Its causes and origins have not been fully understood. Alzheimer’s disease is most common in the elderly who are over 65 years old, and causes a decrease in in mental performance not directly related with age. The disease process causes damage to the cerebral cortex, which in turn leads to trouble with speech, memory or thinking. AD is a condition that develops very slowly and causes problems in everyday life over time. Ultimately, brain atrophy occurs, which is evident during computed tomography or magnetic resonance imaging.

Over the past years, a new hypothesis has emerged in which scientists point to inflammation as the primary cause of the AD. This hypothesis came from an observation that people with RA after use of NSAIDs had an unexpectedly low prevalence of dementia [103]. It is also notice worth that, according to the results published by researchers from the University of Oxford, UK, MTX can reduce risk for dementia, e.g., AD among patients with rheumatoid arthritis. They observed a significant reduction in risk for dementia among patients treated with MTX compared to non-treated patients. The therapeutic effect of MTX probably is caused by a lower dose than that for RA treatment, which is sufficient enough to cross the blood–brain barrier (BBB) to have an effect in AD protection [104]. According to the other research papers, MTX is associated with a lower risk for AD. Based on this knowledge, therapy with MTX shows promise application as a potential treatment for AD [105,106].

#### 3.1.4. Methotrexate-Based Therapy of Other Diseases

Dermatology is another field where MTX is widely and successfully used [6,107,108,109,110]. As recommended, the drug can be orally administered or parenterally by injection and the first effects of MTX treatment usually appear after 2–6 weeks. In accordance with the literature, MTX with beneficial effects has been used in the treatment of other skin diseases, e.g., mycosis fungoides and lichen planus [111,112,113]. MTX has been also used successfully in the treatment of sarcoidosis as an alternative to corticosteroids [68,114,115]. Based on clinical trials, MTX is recommended as a first-line drug for patients with contraindication to corticosteroids because of toxicity or lack of efficiency [116].

Based on American College of Gastroenterology guidelines, MTX is effective and recommended for patients with steroid-dependent Crohn’s disease and for maintenance of remission. The published controlled trials comparing MTX with a placebo show that LDMTX intramuscularly combined with prednisone for 16 weeks or LDMTX monotherapy over 40 weeks, caused greater remission of the disease than the placebo [117,118]. Based on this knowledge, MTX may have a valuable role in selected patients with steroid-refractory or -dependent Crohn’s disease, even if MTX is not indicated by the US Food and Drug Administration (FDA). 

Multiple sclerosis (MS), a potentially disabling disease of the brain and spinal cord, was first described in the 19th century by Jean-Martin Charcot [119]. Most often it concerns young people, with the peak incidence between 20 and 40 years of age, and a slight prevalence of cases in women than men.

MS is a disease where the immune system attacks neurons, oligodendrocytes, and brain immune cells in which the myelin sheath around the nerve cell appendages is damaged, thus causes inability to properly transmit impulses along the nerve pathways in the brain and spinal cord. The term "disseminated" reflects the spread of the pathological process to different places in the nervous system, as well as the interval of change over time. The disease usually has a multi-phase course, with periods of exacerbation and remission [120] 

MTX is an off-label drug for multiple sclerosis treatment however, research has shown that it can reduce relapse rates and slow progression in people with MS, even if the mechanism of action of MTX is still a puzzle. A few trials of MTX have been conducted in multiple sclerosis patients. Goodkin et al. described that LDMTX orally has a slight benefit for MS patients compared to the placebo [19]. LDMTX has been also studied in comparison to IFN β-1α or with a combination of both, demonstrating that the treatment with MTX alone was effective, however combination therapy gives superior results [121,122]. A similar point of view is presented by Stark et al., where the MTX is described as a safe therapeutic option in advanced MS [123].

Information concerning MTX application as non-oncology drug is presented in Table 1.

### 3.2. Methotrexate—the Anti-Tumour Agent

Anti-folates are the first class of anti-metabolites introduced to the clinic approximately 60 years ago. Folate antagonists were among the first antineoplastic agents to be developed [63]. In 1948, the 4-amino derivative of FA (aminopterin) (Figure 1) was used to induce remission in childhood acute lymphoblastic leukaemia (ALL), and the related agent MTX is still an important component of modern cancer treatment [62,124,125]. MTX is FDA approved, alone or with other drugs, for ALL treatment both when it has spread to the central nervous system (CNS) or to prevent it from spreading there, as well as to treat people with other types of haematologic malignancies, e.g., advanced non-Hodgkin lymphoma (NHL) or advanced mycosis fungoides (MF). MTX is approved also to be used in the treatment of many other types of cancer like brain tumours, breast cancer, hepatoma, lung cancer, lymphomas, certain types of head, neck and esophagogastric carcinomas, gastric cancer, osteosarcoma that has not spread to other parts of the body or following primary tumour surgery, prostate and bladder cancers or gestational trophoblastic neoplasia [62,126]. 

However, MTX used in high doses in cancer therapy can also cause very serious, life-threatening side effects highly dependent on the treatment duration, age and condition of the patient. The minimal effective concentration and minimal toxic concentration of MTX is considerably small and the drug has a very narrow therapeutic range [127]. The risk of side effects is greater in patients regularly taking other medicines, especially NSAIDs like aspirin, ibuprofen or naproxen. HDMTX applied in oncological treatment, administrated for a long time, can cause a decrease in the number of blood cells and decrease of the activity of the immune system. HDMTX may cause respiratory failure and liver disease complications (especially in the elderly, obese or diabetic), serious skin reactions, lining damage of mouth, stomach or intestines and even foetal harm or death. Generally, during oncological therapy with HDMTX, the adverse reaction effects happen in more than 1 to 10 people (>10%). Therefore, MTX treatment requires close monitoring of the patient’s health and, if necessary, dose changes, FA or leucovorin (LV, folinic acid) rescue administration or even withdrawal from treatment plan altogether. Nevertheless, HDMTX with LV rescue has been most commonly used in a therapeutic strategy in oncology for almost three decades [126,127].

Concise information concerning MTX application (both alone and as a component of combination regimens) as an anti-tumour agent is presented below and in Table 2.

#### 3.2.1. Methotrexate-Based Therapy of Haematologic Malignancies 

Haematological malignancies (also called as liquid tumours) are a heterogeneous group of blood cancer (neoplastic diseases of the haematopoietic and lymphoid tissues), with clinical behaviour corresponding to leukaemia, lymphoma or myeloma (the subtypes of haematological malignancy according to World Health Organisation (WHO) classification) [124,125]. Leukaemia is defined as a progressive, malignant disease of the blood-forming organs, characterized by distorted proliferation and development of leukocytes and their precursors in the blood and bone marrow. Four subtypes of this malignance can be distinguished: ALL, acute myelogenous leukaemia (AML), chronic lymphocytic leukaemia (CLL) and chronic myelogenous leukaemia (CML). Lymphoma, defined as any malignancy of the lymphoid tissue, is distinguished into two main types: Hodgkin lymphoma and NHL. Both of these are of B-cell origin [239], while almost 90% of lymphomas are extranodal NHL and over 90% of lymphomas are of B-cell origin [124,178].

The most common form of T-cell lymphomas is cutaneous neoplasm MF, also known as Alibert-Bazin syndrome or granuloma fungoides. Multiple myeloma is a cancer of the plasma cells. The malignancy of haematologic neoplasms ranges from relatively indolent to highly aggressive. 

MTX as a cytostatic drug and is widely used to treat many cancers of haematological origin alone or in combination regimens with other anticancer agents (Table 2). The first applications of MTX were used in the 1950s/1960s for childhood ALL treatment [128,167,168,169,170]. Patients were given repeated infusions of HDMTX followed by LV rescue administration. Despite numerous studies conducted in many patients, intravenous (i.v.) HDMTX has not been proved conclusively to be more effective than less toxic and less costly methods based on LDMTX [128]. The promising and effective result of HDMTX application has been described by Sakura et al. (a randomized phase III trial comparing HDMTX therapy with intermediate-dose (IDMTX) therapy) [166]. Adult patients with Philadelphia chromosome negative ALL were randomly assigned to receive therapy containing HDMTX (3 g/m^2^) or IDMTX (0.5 g/m^2^). The estimated 5-year disease-free survival rate was 58% and 32% in patients treated with HDMTX and IDMTX, respectively. 

A deep analysis of efficacy of HDMTX application in the treatment of children with ALL is presented in the paper of Gong et al. [177]. Children with ALL were divided into smaller subgroups according to the subtype of the disease (B- or T-lineage ALL) and disease course. At different time periods after HDMTX infusion, the MTX plasma concentrations and adverse reactions were tested and compared. The obtained results showed that MTX plasma concentration in the B-lineage group was significantly higher than for T-lineage group and the incidence of adverse reactions in the high-risk group was significantly higher than for the moderate- and low-risk groups. HDMTX caused more adverse reactions in B-lineage ALL children compared to T-lineage ones. During clinical application of HDMTX in the treatment of ALL, particular attention was paid to changes in the vital signs of patients and, if necessary, calcium formyltetrahydrofolate or citrovorum factor rescue was administrated.

MTX application in lymphomas are reported in numerous papers. In this work, we selected the most comprehensive articles on MTX-based lymphoma therapy (Table 2) [124,129,171,172,173,174,175,176,179,182,240,241]. Primary central nervous system lymphoma (PCNSL) is a malignant disease of the lymphatic system that accounts for about 2% to 5% of all primary intracranial tumours in immunocompetent patients. It is an aggressive NHL of B-cell origin and can be located in the brain, meninges, eyes, spinal cord, cerebrospinal fluid and intraocular structures [178,182]. Studies of the use of HDMTX as a single agent followed by LV rescue in patients with widespread NHL have demonstrated the effectiveness of this treatment [171,172,173,174,175,176]. Zhu et al. studied the response and adverse effects of intravenous HDMTX administration as a function of age at the time of PCNSL diagnosis [176]. The obtained results showed no significant age-dependent differences in the median progression-free survival (PFS) or overall survival (OS), but higher risk of treatment-related complications in elderly PCNSL patients because of the increased prevalence of comorbidities. MTX in low doses as an anticancer agent is rarely used alone [129,130]. It is usually used in combination regimen with other cytostatics. Wen et al. tested clinical efficacy and toxicity of metronomic administration of LDMTX and thalidomide in relapsed and refractory NHL [136,137]. Generally, the therapy was well tolerated and most patients achieved an objective response (OR).

The most common MTX-based chemotherapy in lymphoma treatment is combination regimen containing HDMTX and other anti-neoplasm preparations [131,132,179,180,181,183,184,185,186,187]. Applications of combination regimen containing HDMTX and cytarabine (Ara-C) for systemic NHL treatment have been described by Bergner et al. [179] and Bokstein et al. [180]. The obtained results indicated a benefit in patients treated with combined HDMTX+Ara-C chemotherapy compared to patients treated with HDMTX alone; however, HDMTX+Ara-C chemotherapy was also less tolerated. Nevertheless, the combination of HDMTX and a high-dose Ara-C regimen is today the backbone of the first line therapy of PCNSL [178]. MTX and temozolomide combination for the elderly suffering from PCNSL treatment was tested by Omuro et al. [181]. Among the patients, a complete response was observed in 55%, and the disease progressed in the other 45%. 

Other HDMTX-based examples of combination regimens used in patients with lymphoma treatment are bleomycin, Adriamycin, Cytoxan and vincristine [172]; rituximab, CTX, doxorubicin, vincristine, prednisolone (R-CHOP) and intrathecal MTX [183]; rituximab, MTX, procarbazine and vincristine (R-MPV) [184]; MTX and rituximab [185,186,187]; rituximab, MTX, Ara-C, dexamethasone (R-MAD) [182]. Generally, outcomes of PCNSL treatment with HDMTX-based regimens remain poor, presumably due to the low ability of MTX to penetrate CNS. In addition, MTX has limited use in cancer chemotherapy because of its dose-dependent adverse effects, such as poor bioavailability, toxicity, low specificity, and drug resistance. The use of combination regimens in most therapies led to an increase in complete response rate and a significant prolongation of PFS compared with the results obtained in HDMTX monotherapy. 

LDMTX alone [131,132,133,134] or in association with other chemotherapeutic agents is recommended for the treatment of primary cutaneous CD30-positive lymphoproliferative disease, as well as alone [131,132,133,134] or in combination with interferons [131,144,145,146,147] recommended for treatment of refractory MF neoplasm. The lateral combination regimen resulted in a very high clinical efficiency and durable response without serious adverse reactions [131,144,145]. In the 1980s, MTX was used to MF treatment in combination chemotherapy containing bleomycin and doxorubicin or along with topical nitrogen mustard [147]. The lateral combination turned out to be quite effective for the treatment of MF advanced stages. Similarly effective for advanced MF treatment was a regimen containing MTX and fluorouracil (5-FU) [189]. MTX administration followed by 5-FU and LV rescue was well tolerated and did not cause any serious side effects. Some years ago, Kuo et al. reported the positive results of the treatment of erythrodermic using LDMTX and UV-B irradiation [148]. The use of MTX in another therapeutic application was presented by Raychaudhury [149]. LDMTX along with radiotherapy including total skin electron beam therapy is recommended in India for intermediate-risk patients with MF.

#### 3.2.2. Central Nervous System—Methotrexate Brain Cancer Therapy and Brain Drug Delivery 

Brain tumours can be divided into two main types: malignant or benign tumours. The limited success of chemotherapy against brain tumours is caused mainly by low CNS drug penetration through the BBB [242]. MTX brain penetration is poor and due to its application at LDMTX is insufficient to achieve therapeutic concentration in the brain [190,191,243,244,245]. In order to attain the desirable MTX concentration in brain cancer, administration of HDMTX is required and current efforts in the field of MTX brain tumour therapy focuses mainly on the design of novel, non-invasive, safe and effective MTX delivery systems for cerebral treatment [199,246,247,248]. An important role in this area play a nanoparticulate (NP) delivery systems investigated with the aim of targeting therapeutics to the brain [199]. Although MTX has been used in medicine for decades, this type of delivery systems began to appear only for several years simultaneously with rapid development in the field of nanomedicine. Many novel approaches of MTX delivery to the CNS are presented below [63,200,201,202,203,204,205,206,207,208,209,210].

Based on the knowledge that some endogenous amino acids are able to cross the BBB, it can be considered that these amino acids can be used as carriers in molecules transported to the brain. Singh et al. designed reversible conjugates MTX-(glutamine)_2_ and -(lysine)_2_ to enhance the brain availability of MTX [190,191]. The biodistribution studies showed much higher accumulation in the brain of both compared to that of free MTX, indicating effective transport of novel conjugates across the BBB and more effective MTX brain penetration.

Corem-Salkmon, in the frame of novel approach to deliver drugs directly into brain tumours, presented the application of biodegradable magnetic nanoparticles (NPs) with MTX, conjugated with the human serum albumin (HSA) [192,193]. Studies on formulations of MTX-HSA have been described also by Wosikowski et al. [194] and Burger et al. [195]. The study results showed that this approach can alter MTX pharmacokinetic, enhance tumour targeting, reduce toxicity, overcome drug-resistance mechanisms and extend the clearance time of the conjugate. Kohler et al. designed, synthesized and studied magnetic iron oxide nanoparticles conjugated with MTX [196]. This preparation can be used both as a diagnostic agent in magnetic resonance imaging and as a carrier in controlled drug delivery. Gao and Jiang studied the transport ability of MTX-loaded polyacrylate nanoparticles [197]. MTX NPs, especially of diameter below 100 nm, had the ability to cross the BBB. Application of such conjugates significantly improved MTX level in both brain tissues and cerebrospinal fluid (CSF). Trapani et al. described the synthesis and physicochemical properties of formulation of MTX with chitosan NPs [198]. These nanocarriers seem to act as MTX reservoirs, allowing slow drug release in the tumour area, once the BBB is crossed [192,198]. Another MTX-loaded NP described in the literature is based on the biodegradable polymers: polyesters of (poly(lactic acid) (PLA) and poly(d,l-lactide-co-glycolide) (PLGA) [199,249]. Both these polymers are preferred as components for formulations indicated for brain drug delivery as the non-toxic products of their autocatalytic hydrolysis are degraded to CO_2_ and H_2_O. Studies of physicochemical and biological properties of MTX-polymeric NPs showed that these MTX-loaded NPs improved MTX delivery to the target, what increased therapeutic effect and reduced serious adverse effects on normal tissues such as kidney and liver. An MTX-based polymeric delivery system can be also prepared from proteins (gelatine, albumin) [199,200,201,202,203,204,205,206]. Narayani and Rao presented biodegradable hydrophilic MTX-gelatine microspheres [201,202]. The rate of MTX release from microspheres decreased with increase in the particle size of the microspheres and was faster in gastric fluid than that in intestinal fluid. Another MTX formulation in the form of gelatine microspheres was studied by Pica et al. [203,204]. These preparations, intended for slow drug release directly within the tumour, have been designed to minimize the systemic toxic effects of MTX and to overcome tumour resistance. However, contrary to expectations, the molecules of free MTX were released more efficiently from microspheres containing less tethered drugs. Hydrophilic biodegradable gelatine microspheres with covalently attached MTX provided prolonged drug release and increased anti-tumour activity. Silica-gelatine hybrid aerogels [205] are excellent candidates for controlled MTX delivery. Application of these aerosols can provide the same therapeutic effect inside the tumour site as free MTX, but without any adverse effects. 

A promising platform applied in nanomedicine for the design of new drug delivery systems is dendrimers [206,207,208,209]. Application of MTX formulations built on generation 5 (G5) acetylated poly(amidoamine) (PAMAM) dendrimers have been tested and described by several research groups [206,207,208,209]. Wu et al. described synthesis as well physicochemical and biological characterization of multi-component conjugate C225-G5-MTX containing monoclonal antibody cetuximab (also called as C225), covalently linked to G5 PAMAM dendrimer, and previously coupled with a cytotoxic drug MTX [208]. This formulation is an effective way to deliver the drug to the target tissue, and can also increase the therapeutic index of MTX. The formulation G5-MTX containing MTX molecules tethered in the PAMAM G5 dendrimer makes MTX action about 10 times stronger and much less toxic than in the case of drug alone application [209]. Conjugates of MTX molecules and the fifth-generation PAMAM dendrimer, G5(MTX)_n_, where n = 0÷12, were synthesized and studied by Wong et al. [62]. Based on the obtained results, the authors concluded that these conjugates bind to FRs three to four orders of magnitude stronger than free MTX and that the drug molecules tethered in dendrimer can still inhibit a human DHFR enzyme as potently as free MTX does; however, this potency strongly depends on the length of applied in the syntheses linkers between dendrimer and MTX molecules. The studies have also showed the effectiveness of G5(MTX)_n_ conjugates application in a tumour-targeting nanodelivery strategy. Another dendrimer-based MTX conjugate has been described by Dhanikula and Hildgen [210]. In order to evaluate the potential application of these drug delivery systems, the authors studied the efficiency of MTX encapsulation and release from polyester-co-polyether (PEPE) dendrimers. Riebeseel et al. designed and studied hybrid conjugates (MTX-PEG) of MTX and poly(ethylene glycol)s (PEGs) [211]. PEGs are potential drug carriers for improving the therapeutic index and delivery system of anticancer agents. All of the MTX-PEG conjugates inhibited DHFR in a similar manner to free MTX, but they were slightly less active than free MTX. The level of cytotoxicity of these preparations decreased with increasing sizes of MTX-PEG polymer conjugates.

It is also worth mentioning that the MTX-EPR conjugate combines two significant anticancer strategies: combinatory therapy and targeted therapy [212]. The MTX-EPR hybrid molecule contains two anticancer drugs, MTX and epirubicin (EPR), and each of them is characterized with a different mechanism of action. Co-administration of these two drugs in combination therapy causes the appearance of a synergistic effect. In addition, the cytotoxic effect of such a hybrid molecule is lower than in the case when both chemotherapeutics are used in therapy, but they are not administrated simultaneously.

Concise information concerning MTX-based therapy of CNS are presented in Table 2.

#### 3.2.3. Methotrexate-Based Therapy of Head and Neck 

Head and neck squamous cell carcinoma (HNSCC), cancer that arises in the nasal cavity, sinuses, lips, mouth, salivary glands, throat or larynx, is the 10th most common tumour worldwide. These sarcomas are rare cancers, and due to their variable clinical behaviour, location in close proximity to important vital structures and diversity of histologic subtypes, these neoplasms are difficult to diagnosis and effective treatment. Nowadays, in the era of personalized medicine, head and neck carcinomas belong to a critical oncological topic. Conventional HNSCC chemotherapy consists of drugs administration in cycles near or at the maximum tolerated dose, followed by an appropriate drug-free period allowing the patient to recover from acute drug toxicities. MTX, particularly in a high dose with LV rescue, is one of the main cytostatics used in head and neck carcinoma therapy (Table 2). It is used alone or in a combination regimen with other drugs; however, head and neck cancer therapy also requires radiation therapy and/or surgical tumour removal. MTX in different treatment regimens has been used in HNSCC therapy for several decades [138,139,188,250,251,252,253,254,255,256,257]. The use of MTX alone is described in numerous works [139,211,250,253,254,258,259,260,261,262]. MTX in different combination regimens is considered as a first-line drug indicated for palliative care in the case of head and neck carcinomas [261,262]. However, monitoring the patient condition is particularly important here, as patients are often elderly people with a multitude of comorbidities.

MTX alone or in combination with other drugs is used in metronomic chemotherapy in metastatic HNSCC patients [140,141,263,264]. Pai et al. described the application of LDMTX and celecoxib in locally advanced (stage III and IV) oral cancers [141]. The treatment was carried out twice before the surgery and during the perioperative period. This turned out to be more effective compared to cisplatin (PDD, cis-diammindichloridoplatin) reference chemotherapy. The authors documented in their work the advisability of using MTX in the neoadjuvant and adjuvant setting in advanced HNSCC. Mateen et al. described metronomic chemotherapy procedure for head and neck cancer therapy of oral LDMTX and capecitabine for at least a period of six months after chemoradiotherapy [143]. Based on the recorded results, the authors concluded that this regimen is an effective treatment option for recurrent HNSCC.

MTX is used from the past to now as a component in many combination regimens. Bertino et al. [203] tested the efficiency of the combination of MTX and bleomycin; these attempts have been discontinued due to high side effect toxicity. Next, the authors tested the effects of simultaneous administration of MTX and 5-FU. Application of HDMTX with bacilli Calmette Guerin vaccine described by Buechler et al. [188] provided only a temporary response. The combination regimen of MTX and LV with cyclophosphamide and cytosine arabinoside (MLCC) is presented in work of DeConti et al. [255]. Application of this regimen compared to MTX with LV resulted in more hematologic toxicity. The aim of the study conducted by Hong et al. was to compare MTX therapy versus PDD in patients with recurrent HNSCC [256]. The obtained results showed comparable effects of both cytostatics, however MTX proved to be better tolerated. In addition, comparison of the efficiency of MTX and a combination regimen containing a high-dose of PDD, Oncovin (vincristine) and bleomycin (COB) is discussed in the paper of Drelichman et al. [257]. Therapeutic effects of both sets turned out to be comparable, however nausea and vomiting were the more common side effects of COB, while hematologic toxicity was more frequent and more severe in the case of MTX application. Another therapeutic kit containing MTX, LV, PDD and 5-FU was tested by Chang et al. [139]. This regimen showed total safety profile and improved survival in patients with metastatic/recurrent HNSCC. Kushwaha et al. compared therapeutic effect observed in recurrent HNSCC treatment of three different regimens: gefitinib, MTX or MTX combined with 5-FU [140]. The authors observed comparable median survival rate in all cases, however a different toxicity profile. Adverse toxic effects oral mucositis, nausea, vomiting and haematological toxicity were recorded in the case of regimens of MTX and MTX with 5-FU while cutaneous rash, diarrhoea and oral mucositis were observed in the case of gefitinib. MTX or afatinib application in the treatment of HNSCC was studied and evaluated by Guo et al. [135]. Afatinib proved to give a significant improvement in patient PFS, slightly better than MTX. Very optimistic test results were obtained in the treatment of advanced/recurrent HNSCC patients using metronomic chemotherapy of MTX and celecoxib combination [142]. Celecoxib enhanced the therapeutic efficiency of MTX, treatment was well tolerated and provided good pain control and improves quality of life.

#### 3.2.4. Methotrexate-Based Therapy of Breast Cancer

Breast cancer is the most frequently diagnosed life-threatening cancers in women and the main reason of death among women with cancer. MTX is used (alone or in combination) to treat both early-stage (as well after surgery and other treatments) and advanced-stage breast cancer. Some selected application of MTX in breast cancer therapy are presented below [196,213,214,215,216,217,218,219,220,221,244,265,266,267], Table 2. Martin et al. described numerous combination regimens applied in metastatic breast cancer, such as CTX, MTX, 5-FU (CMF), 5-FU, Adriamycin, CTX (FAC) and MTX in combination with tegafur and uracil followed by LV rescue (MUL) [213,214]. The authors demonstrated the superiority of MUL regimen compared to other ones. This regimen stopped disease progression in nearly 75% of patients. MUL chemotherapy turned out to be active and well tolerated and the toxicity was mainly gastrointestinal. Application of the regimen containing MTX and CTX in patients with metastatic breast cancer is described by Colleoni et al. [218]. Continuous administration of both at very low doses over a long period of time without treatment interruptions (typical for anti-angiogenic therapy) was well tolerated. The same research group presented the benefit of application of combination regimen CMF followed by tamoxifen or tamoxifen alone, as adjuvant chemotherapy in women with lymph node-negative breast cancer [266]. Based on experimental data, the authors showed that the classical CMF regimen is safe and plays a key role in the response to therapy in early breast cancer [215,216]. Cocconi et al. compared CMF regimen with the modified CMF method they proposed. The new regimen, CMFEV, contained CMF components and epirubicin or vincristine, rotationally [216]. The study demonstrated superiority of the CMFEV regimen over CMF. CMFEV showed greater anticancer activity in premenopausal patients and more effective in terms of long-term outcomes. Leone et al. described applications in breast cancer therapy for the regimens CMF and FAC [217]. The aim of the study was to describe the long-term results of these regimens application in neoadjuvant chemotherapy in stage III breast cancer patients. Both regimens were well tolerated and showed similar objective response (OR), long-term toxicity, disease-free survival (DFS), and overall survival (OS) rate at 16 years. Wu, C. E. et al. described a CMF regimen application in adjuvant chemotherapy of triple-negative breast cancer (TNBC), characterized with negative tests for oestrogen receptor (ER), progesterone receptor (PR), and human epidermal growth factor receptor-2 (HER2) [267]. This kind of therapy turned out to be effective in reducing recurrence amount and prolong DFS in patients with node negative TNBC, particularly in the case of tumours more than 2 cm or tumours after partial mastectomy. Fukuda et al. described a retrospective analysis of the efficacy of drugs mitomycin C and MTX in HER2-negative patients with metastatic breast cancer [219]. They described also the combination chemotherapy of mitomycin C and MTX (MM) in patients previously treated with aggressive anthracyclines, taxanes, capecitabine and vinorelin (ATCV) regimens. Under these conditions, MM chemotherapy proved to be effective and tolerated, however, only in patients who kept good performance status and bone marrow function even after multiple chemotherapy regimens. The combined effects of MTX and 5-aminoimidazole-4-carboxamide riboside (AICAR) on tumour cell proliferation were the object of research conducted by Cheng et al. [220]. In the frame of these researches, the influence of MTX on AICAR anticancer action and pharmacokinetics in human breast cancer as well as concentration of active AICAR metabolites in plasma and tumours were studied. The authors recorded a synergistic effect of MTX and AICAR combination on the inhibition of tumour cell proliferation—the combination therapy displayed more potent cytotoxicity against tumour cells than either drug alone. The synergistic effect of anti-tumour action was observed also in TNBC treatment reported by Wu, C. W. et al. [221]. They proposed the simultaneous use of MTX and a low dose of vitamin C. Both drugs taken separately do not have the ability to inhibit the growth of TNBC cells, however, the combination of vitamin C and MTX had synergistic anti-proliferative and cytotoxic effects on TNBC cells.

Breast cancer usually spreads to the CNS inducing commonly leptomeningeal metastasis. Kapke et al. determined that systemic intravenously administration of HDMTX, improves quality of life and provides durable remissions for leptomeningeal metastasis of breast cancer [244]. Application of MTX in a form of iron oxide NPs for human breast cancer therapy is given in the paper of Kohler et al. [196]. The release of MTX inside the tumour cells occurred as a result of the drug cleaved from the NPs under intracellular low pH conditions in cancer cells. Kumaki et al. described the case of two breast cancer patients with a history of LDMTX treatment for RA, who had histological findings similar to those seen after neoadjuvant chemotherapy [268]. The authors showed that oral LDMTX induced histological results similar to those seen in HDMTX chemotherapy, such as metronomic chemotherapy.

#### 3.2.5. Methotrexate-Based Therapy of Lung Cancer

Lung cancer is one of the most common types of cancer, and according to current knowledge, the occurrence of this tumour is closely related to smoking [269]. Worldwide, over 1 million people are diagnosed with lung cancer each year. There are two main categories of lung cancer: small cell lung cancer (SCLC) and non-small cell lung cancer (NSCLC). NSCLC (any type of epithelial lung cancer) accounts for about 85% of all lung cancers. Lung cancers are primarily treated by surgical resection, although chemotherapy (among others with MTX) is also used both pre-operatively (neoadjuvant chemotherapy) and post-operatively (adjuvant chemotherapy). In general, NSCLC is relatively insensitive to chemotherapy, while SCLC has a faster growth rate, shorter remission times and previous metastases to various locations in the body (e.g., cervical, prostate, gastrointestinal tract). According to the literature data, MTX is relatively rarely used in lung cancer treatment, however the examples of its application in lung cancer therapy have been described in several papers (Table 2).

Vincent et al. described and compared the use of a single drug MTX and five different combination regimens in SCLC chemotherapy [151]. The regimens always contained the drugs MTX, CTX and lomustine and one of the following chemotherapeutics etoposide, Adriamycin or vincristine. The most effective regimen was a set containing all of the above-mentioned drugs followed by radiation therapy to the chest. The therapy was well tolerated by patients, and provided a high objective tumour response rate. Handle et al. presented SCLC therapy based on two regimens—HDMTX or LDMTX with LV rescue in combination with CTX, doxorubicin and vincristine cycles alternating with etoposide, vincristine and hexamethylmelamine cycles [152]. In both approaches, the recorded response rates, median survival and overall survival were comparable, but adverse effects were more common in the case of HDMTX application. Overall, the HDMTX-based approach was less effective and associated with a higher cost and toxicity. Therapy of NSCLC with HDMTX followed by citrovorum rescue, mechlorethamine and procarbazine was studied by Bhasim et al. [222]. Although HDMTX followed by citrovorum rescue was relatively well tolerated, the addition of these two medications failed to improve the tumour response rate and increased toxicity. Another combination regimen containing LDMTX or HDMTX and drugs vincristine (Oncovin), doxorubicin and CTX (MOAC) for SCLC treatment are presented in the work of Neijstrom et al. [153]. Although HDMTX was able to achieve potentially therapeutic concentrations of MTX in CSF, the results of the study were not conclusive. There were no statistically significant differences in response rate or survival between these two approaches and there was no basis to conclude that HDMTX could prevent brain metastasis or the appearance of other CNS diseases. Interactions of LDMTX and PDD in NSCLC treatment were studied by Preiss et al. [154] and results here were not conclusive as well. Contrary to expectations, the authors observed no significant change in the kinetics of PDD in plasma after co-administration of MTX. After combined drugs administration, no signs of nephrotoxicity were observed and the overall toxic effects were on a mild level. Malzyner et al. studied, evaluated and compared numerous combination regimens used in advanced NSCLC therapy [155]. The authors described two pairs of sets: the MVP set containing mitomycin C, vinblastin and PDD and the M-MVP set containing additional MTX, the MNP set containing mitomycin C, vinorelbine and PDD, and the M-MNP set containing additional, as before, MTX. Based on the obtained results, Malzyner et al. reported superiority of the M-MVP regimen compared to MVP and the M-MNP compared to M-MVP. The M-MNP regimen, characterized with high positive response rate, as well as mild and manageable toxicity in patients.

Lung metastasis is the most crucial event affecting the treatment of osteosarcoma and is strongly dependent on tumour angiogenesis. With this in mind, Tomoda et al. investigated the inhibitory effect of MTX on lung metastasis, and on angiogenesis associated with the early stage of tumour formation [270]. The study was performed using a rat osteosarcoma S-SLM cell line with high metastatic potential. The obtained results showed that LDMTX, through its anti-angiogenic activity, significantly suppressed lung metastasis. In their work, the authors also drew attention to the fact that, although HDMTX is commonly used for the treatment of osteosarcoma, the conventional MTX dose in this therapy is ineffective. In the experiments performed in the frame of this work, low MTX doses could not inhibit the growth of tumours and kill osteosarcoma cells; however, they had a potency to suppress the lung metastasis process. Based on these facts, the authors concluded that MTX mechanisms of anti-angiogenic action and cytotoxicity are different.

Chen et al. designed and tested a new formulation MTX-PMX-PCNPs in which two different drugs MTX and pemetrexed (PMX, novel anti-folate with indication for lung cancer and malignant pleural mesothelioma treatment) were bonded with PEGylated chitosan nanoparticles (PCNPs) [223]. Drug release profiles of MTX and PMX were investigated using LLC and A549 cell lines. The obtained experimental results showed a stronger overall effect of the MTX-PMX-PCNP conjugate on the decreasing cell proliferation than the combined effects of MTX and PMX used separately, indicat that MTX and PMX had synergistic anti-tumour activity. The IC_50_ value of the formulation was significantly lower than those of free drugs. The use of MTX and a potent tubulin-binding anti-tumoural drug pretubulysin, alone and in the form of a combined regimen, is described by Kern et al. [224] The authors demonstrated the benefits of using the drug regimen and indicated that it was a promising therapeutic approach for various types of cancer, among other in lung cancer therapy. Sekimura et al. presented a case of primary lung cancer with MTX-associated lymphoproliferative disorder [271] which occurred in a patient receiving oral MTX and prednisolone for RA for about 15 years. Lung cancer therapy in such cases requires discontinuation of MTX, accurate diagnosis and initiation of other kind of treatments. Yan et al. noted the influence of acetylsalicylic acid (ASA) on the therapeutic anticancer activity of MTX [225]. ASA is a non-selective cyclooxygenase inhibitor that takes part in the treatment of inflammatory conditions such as, for example, RA. The anticancer activity of MTX and ASA were tested both alone and in combination, using two cell lines of human NSCLC, CL1-0 and A549. Based on experimental results the authors demonstrated that ASA suppresses the therapeutic efficacy of MTX in human lung cancer treatment via preserving cancer cell proliferation and survival. They concluded that the efficacy of MTX anticancer therapy can be decreased by ASA and, due to that application, these two medications together should be avoided. The novel and effective way of MTX pulmonary delivery in case of lung cancer treatment has been designed by Abdelrady et al. [226]. The authors designed and tested the application of MTX-gelatine nanoparticles obtained in different MTX loading techniques. The new preparation was well tolerated and had low toxicity, but most importantly, a satisfactory therapeutic effect was achieved with four times lower doses of MTX than in the case of standard MTX-based lung cancer treatment.

#### 3.2.6. Methotrexate-Based Therapy of Prostate and Bladder Cancers

Urothelial cancer (UC) is the fifth most common malignancy both in men and women (the male to female occurrence is 3:1). Approximately, more than 400,000 new cases are being recorded annually and approximately 150,000 die from this disease, worldwide. Bladder cancer is a common urologic cancer that has the highest recurrence rate of any malignancy. MTX has been used for bladder cancer treatment since the 1970s to the 1980s of the last century [10,156,157,158,159,160,161,162,163,227,228,272]. It was used both alone [10,156,158,228] and in combination regimens with such antineoplastic agents as PDD, vinblastine, Adriamycin, 5-FU, vincristine, bleomycin, mitomycin C or cyclophosphamide [10,159,160,161,162,272]. MTX used as a single agent is one of the most active drugs [161].

One of the first applications of MTX-based treatment of locally advanced or metastatic transitional cell carcinoma (TCC) of the bladder are presented in the works of Hall et al. [156] and Turner et al. [158]. Hall’s research showed objective evidence of tumour regression in 26% of patients and several other patients gained relief of disease symptoms, but in these there was no clear evidence of tumour regression. Occurring toxic side effects associated with MTX application caused treatment interruption for a time in most patients [156]. Turner et al. described studies in which MTX was used in patients with advanced bladder cancer that did not respond to previously applied radiotherapy or had recurred after radiotherapy, and patients with metastatic disease. Patients were treated with LDMTX i.v. with or without LV rescue [158]. The obtained results show high effectiveness of these therapies, while the intensity of the side-effects was related with LV rescue. Generally, the medical experiments described above showed that PDD therapy is of value in the management of advanced bladder cancer. Natale et al. reported results of patients suffering from transitional urothelial tract tumours treated with LDMTX i.v. with or without citrovorum factor rescue [10]. Based on the obtained results, authors concluded that MTX in the treatment of patients with advanced urinary bladder cancer is as active as PDD. In another therapeutic approach, patients with advanced primary bladder cancer and metastatic lesions were treated using combination regimen consisting of MTX, 5-FU, vincristine, bleomycin and mitomycin C [227]. Complete and partial responses were recorded in the case of 9% and 23% of patients, respectively. HDMTX-based treatment with LV rescue of patients with TCC of the bladder has been described by Hall et al. [228]. The authors concluded that transurethral resection followed by HDMTX application may offer an effective alternative to radiotherapy or cystectomy for many patients with invasive bladder cancer. Kaye et al. described the results of the patients with advanced bladder cancer using MTX alone or in combination with PDD [159]. The obtained results showed that patients with satisfactory renal function can receive low doses of both drugs simultaneously without risk of enhanced toxicity. MTX is used as a component of numerous anticancer combination regimens [160,161,162,163,164], while the most commonly used in the advanced bladder carcinoma are MTX-vinblastine-Adriamycin-cisplatin (MVAC), HDMTX-vinblastine-Adriamycin-cisplatin (HDMVAC) and cisplatin-MTX-vinblastine (CMV). Since 1990, the MVAC and HDMVAC have been considered as standard first-line therapies in this indication [162]. MVAC showed superior effectiveness than single agent PDD, despite increased toxicity [162,163,164,165].

#### 3.2.7. Methotrexate-Based Therapy of Osteosarcoma

Osteosarcoma is the most common histological form of primary bone cancer that is characterized histologically by the production of osteoid by malignant cells [165,273]. This neoplasm is most prevalent in children and young adults (it accounts for about 3.4% of all childhood cancers and 56% of malignant bone tumours in children). It is characterized by local infiltration and early, distant, hematogenous metastasis [236]. HDMTX with LV used alone or in different regimens exhibit significant activity against relapsed and metastatic osteosarcoma. In the work of Jaffe et al., four combination regimens of HDMTX-LV with doxorubicin, PDD, ifosfamide or CTX utilized in osteosarcoma treatment are presented [229]. As a main adverse effect, the authors mention renal failure caused probably by MTX precipitation in the renal tubules. Other toxic effects were gastrointestinal, hematologic, and hepatic dysfunction. Xu, M. et al. analysed the clinical efficacy of the cisplatin-ifosfamide-doxorubicin (DIA) regimen in osteosarcoma treatment and compared this therapy with HDMTX [274]. DIA therapy turned out to be better tolerated. Bielack et al. described the research conducted by European and American Osteosarcoma Study Group. They tested the final effect of therapy in which the treatment process was extended by the additional stage including treatment with MAP regimen (containing MTX, doxorubicin and PDD) enriched with the PEGylated formulation of IFN-α-2b [230]. The obtained result was not conclusive, because the slightly more effective final result of the extended therapy was not statistically different from the effect of the therapy without this additional stage. The problem of appropriate dose indication for MTX as a component of a multidrug regimen for the osteosarcoma treatment was tested by Krailo et al. In the frame of a randomized clinical trial in postoperative chemotherapy in the treatment of childhood osteosarcoma, two regimens were tested and compared: Adriamycin and vincristine combined with HDMTX or a moderate MTX dose. Both therapies showed similar effectiveness [231], which allows the conclusion that a moderate dose of MTX is sufficient in postoperative multidrug adjuvant chemotherapy of osteosarcoma. Mayers et al. investigated the therapeutic effect resulting from the addition of muramyl tripeptide encapsulated in liposomes to a HDMTX, PDD, isofosfamide and doxorubicin regimen [232]. The authors observed that the addition of tripeptide, in practice, did not enhance event-free survival. Crews et al. demonstrated the relation of MTX pharmacokinetics between patient age and outcome in different combination therapies of osteosarcoma [13]. After the HDMTX administration, higher mean serum concentrations, higher exposures, and lower mean clearance of MTX were associated with poorer therapy outcomes. The safety and efficacy of osteosarcoma therapy conducted with HDMTX in adolescents compared with young adults were tested by Wippel et al. [275]. The obtained results showed that young adult patients (aged 19–38) are more likely to experience a delay in MTX clearance when compared to adolescents (aged 7–18) despite using the similar supportive rescue. Application of HDMTX in osteosarcoma neoadjuvant chemotherapy used in the frame of a randomized cooperative trial is presented in the work of Winkler et al. [233]. Two groups of patients were treated with two approaches: the first one was based on preoperative chemotherapy containing HDMTX applied with the triple drug combination of bleomycin, CTX and dactinomycin (BCD) and postoperative salvage treatment with doxorubicin and PDD, and the second, preoperative chemotherapy with doxorubicin-PDD or doxorubicin-BCD system application and postoperative treatment with ifosfamide. The histologic response rate was significantly lower in the case of preoperative chemotherapy containing HDMTX (first approach); however, the four-year metastasis-free survival rate was improved in the case of using doxorubicin-PDD in preoperative chemotherapy (the second approach).

HDMTX-based neoadjuvant chemotherapy for localized osteosarcoma treatment has been described by Bacci et al. [234,235]. The obtained results showed that application of aggressive neoadjuvant chemotherapy: HDMTX i.v. followed by PDD and doxorubicin was allowed to carry more than 60% of non-metastatic osteosarcoma to the extremities, and in about of 80% of patients it allowed the avoidance of an amputation. Ifosfamide and etoposide seem to be indicated in patients who did not respond to preoperative chemotherapy good enough. Combination of MTX and doxorubicin, described by Geller et al. [236], ensured disease-free survival in about 60% of patients. Frequently used in the past combination of bleomycin, CTX and dactinomycin (BCD) has been replaced by a regimen of MTX and Adriamycin. Systemic chemotherapy consisting of HDMTX, PDD, and doxorubicin allowed a cure in approximately 70% of patients with localized osteosarcoma. Unfortunately, a long-term survival rate in patients with metastases was here less than 20%. A new combination containing HDMTX and trimetrexate (a structural analogue of MTX) was tested in the treatment of patients with recurrent osteosarcoma. Another new regimen mentioned in this work was the set containing zoledronic acid in combination with HDMTX, PDD, doxorubicin, ifosfamide, and etoposide, especially indicated for patients with newly diagnosed metastatic osteosarcoma. In the work, the mechanisms responsible for MTX resistance were presented. MTX resistance in osteosarcoma treatment and the ways it is overcome are discussed also in papers of Guo et al. [276], Wang et al. [277] and Xu et al. [278]. Xu et al. proposed a relatively novel method to overcome the MTX resistance in osteosarcoma. The authors used the capacity of microRNAs (miRNAs) to regulate diverse biological processes, including drug resistance. miRNAs are small non-coding RNAs implicated in a growing number of human diseases, including osteosarcoma [279]. Based on obtained results Xu et al. determined that miR-29 family can inhibit MTX resistance and induce cell apoptosis in osteosarcoma. In the end, targeting the miR-29 family might provide to overcome the HDMTX-induced cytotoxicity in osteosarcoma treatment.

#### 3.2.8. Methotrexate-Based Therapy of Gestational Trophoblastic Disease

Gestational trophoblastic neoplasia (GTN) is a pregnancy-related cancer. These tumours are formed when the cells of the womb in the process of placenta formation begin to proliferate in an uncontrolled way. The most invasive neoplasm belonging to this group cancer is choriocarcinoma characterized with quickly haematogenous spreading (mainly to the lungs, brain and liver), but also with high sensitivity to chemotherapy and a very good prognosis. 

In the work of Abräo et al., three treatment regimens—MTX, dactinomycin (ACT) and the combination of MTX and ACT (MACT) as first-line chemotherapy in GTN—are described and compared [237]. The determined rates of total remission were 69%, 61% and 79%, respectively, in the MTX, ACT and MACT regimens, and the adverse reaction rates 29%, 19% and 63% in MTX, ACT and MACT, respectively. As one can see, the most effective therapeutic effect was obtained in the MACT scheme, but in this case the toxic side effects were also the highest. The most optimal agent tested under these studies seems to be ACT. Lurain et al. analysed the efficacy and toxicity of MTX-based chemotherapy in patients with non-metastatic gestational trophoblastic disease [280]. The subsequent stages of therapy included MTX monotherapy, ACT monotherapy, combined chemotherapy and finally hysterectomy. The therapy was well tolerated (the most common side effect was severe stomatitis) and brought healing to all the patients. Based on study results, the authors determined that high human chorionic gonadotropin (hCG) levels are a major factor associated with MTX resistance development. The same GTN therapy profile was applied in the studies described by Chapman-Davis et al. [281], similar to that of Lurain et al. [280]. The deep analysis of single-agent MTX-based chemotherapies performed on nearly 1000 patients are described in the work of Couder et al. [282]. The studies included the tumour treatments in patients with low-risk GTN, who, according to the FIGO classification, were indicated for MTX chemotherapy. Based on the analysed data, the authors concluded that the number of chemotherapy courses needed to normalize the hCG level may be a predictive factor of tumour recurrence—in the patients needing more than four MTX courses, the risk of relapse was definitely higher. The paper of Lawrie et al. [283] presents results of seven randomised controlled trials performed in large group of patients with low risk GTN. Each group consisted of about 100 patients, and in chemotherapy procedures single agent MTX or ACT were applied (both drugs in different doses and in different time intervals were administrated). In general, both therapies were highly effective and well tolerated; however, the authors suggested a slight therapeutic advantage of ACT chemotherapy over the MTX one. Two therapies applied in patients with high-risk GTN are described in the work of Han et al. [238]. The therapies were conducted using the following regimens: HDMTX-etoposide and etoposide-cisplatin in combination with etoposide-MTX-dactinomycin (EP-EMA). The first preliminary studies performed by these researchers showed a better toxicity profile of HDMTX-etoposide than EP-EMA. According to the authors, the HDMTX-etoposide regimen can be indicated for patients with high-risk GTN. The works of May et al., Aminimoghaddam et al. and Alazzam et al. present chemotherapeutic management of women with all stages of GTN as well as with recurrent disease [284,285,286]. In these papers, besides the information about monotherapies conducted with single drugs of, for example, MTX, dactinomycin, etoposide, cisplatin, there are also described application and effectiveness of different regimens. Based on the study results, the authors indicate MTX or ACT as first-line single drugs for low-risk GTN treating and etoposide, MTX, dactinomycin, CTX and vincristine as first-line regimen for high-risk GTN therapy. Nevertheless, the choice of therapy must always take into account the patient’s physical condition, drug tolerance, drug resistance, adverse reactions and the stage of the tumour. The article of Skubisz et al. presents MTX and its mechanism of action, as well its application in GTN and ectopic pregnancy treatment [287]. The authors classify chemotherapy of patients with GTN into two groups, of low risk and high risk, depending on Prognostic Index Score (PIS) determined by WHO. Low-risk women are highly likely to respond to single-agent chemotherapy with MTX or dactinomycin. MTX followed by LV is a first-line treatment in women with low risk GTN. High-risk GTN is defined as stage IV disease or stage II-III disease with a PIS index of seven or higher and in these cases multiagent chemotherapy is needed. Combination regimen of etoposide, MTX, dactinomycin, CTX and vincristine is the first-line regimen for high risk GTN and it is a relatively well-tolerated regimen. The work of Agarval et al. concerns the validation of the uterine artery pulsatility index (UAPI, a measure of tumour vascularity) as a useful indicator for predicting resistance to MTX chemotherapy [288]. A similar issue is also the subject of the works of Secki et al. [289] and Hussain et al. [290]. Stratification of patients of low- or high-risk GTN for single-agent chemotherapy (with MTX-LV or ACT) or multi-agents chemotherapy, respectively, is based on the assessment of the CXH and FIGO indexes. However, this evaluation does not take into account the effect of MTX resistance on the effectiveness of therapy, which is very important especially in the case of low-risk GTN patients indicated for MTX-LV therapy [291,292]. The authors’ findings showed that the UAPI index can be a non-invasive marker of tumour vascularity and MTX resistance, and a useful tool for the indication of appropriate chemotherapy. A low UAPI index value indicates for tumour formation, and increasing the UAPI index value in patients with GTN indicates emerging MTX resistance.

## 4. Conclusions

In the present review, we take an overview of present state knowledge as regards to MTX, a structural analogue of FA, which is one of the most effective and extensively used drugs for treating severe and resistant forms of neurological and autoimmune diseases or many kinds of cancers. We have carefully overviewed the scientific literature (articles, trial studies and guidelines by FDA and EMA) to describe MTX’s action mechanisms and its applications. MTX is already a well-known DMARD and chemotherapy agent. It has been used in low doses as an effective drug in RA for many years and in high doses in the therapy of various types of cancer for decades.

Application of MTX in high doses in cancer therapies is associated usually with serious side effects, even life-threatening for patients. However, prolonged use of MTX in low doses can also result in occurrence of serious side effects. There are numerous reports in the literature concerning MTX-induced neoplasms (mainly lymphomas [293,294,295,296,297,298] and lung cancer [271,299], but also breast cancer [268], hepatocarcinoma [300], acute leukemia [301]) in patients with RA or psoriasis treated with LDMTX administered continuously and gradually for a long period of time.

Relatively recently, reports have appeared in the literature about the application of MTX in the treatment of neurological diseases MG [302] and AD (as a drug potentially able to reduce the risk of dementia) [103,104,105,106]. However, these medical experiments have been conducted on small population sizes and, up to now, without significant results, and more detailed studies are needed for determining the conclusive effect of MTX with regards to these diseases.

Nevertheless, based on the literature data demonstrated in our review, one can conclude that MTX therapy provides sustained clinical benefits with controlled and minimized side effects (using recommended rescue), even in off-label applications. In summary, MTX has been used over 60 years to treat variety of autoimmune disorders with the same or greater efficacy than most other synthetic DMARDs, e.g., AZT, and is also used alone or in combination with other anticancer agents in different types of cancer treatments and still remains of great interest to researchers all over the world.

## Figures and Tables

**Figure 1 ijms-21-03483-f001:**
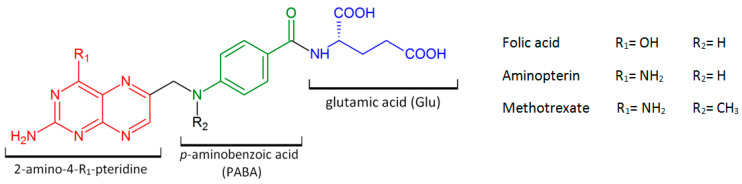
Structure of folic acid and its derivatives - aminopterin and methotrexate.

**Figure 2 ijms-21-03483-f002:**
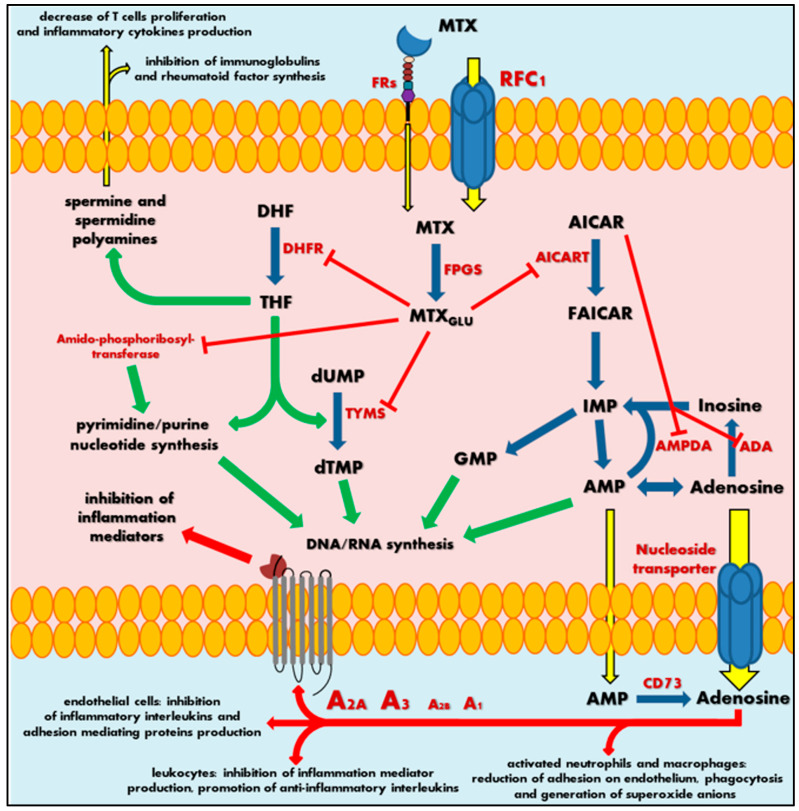
Scheme of mechanism of methotrexate action. MTX—methotrexate; MTX_Glu_—polyglutamated methotrexate; FRs—folate receptors; RFC_1_—reduced folate carrier 1; FPGS—folylpolyglutamyl synthase; DHF—dihydrofolate; DHFR—dihydrofolate reductase; THF—tetrahydrofolate; dUMP—deoxyuridine monophosphate; TYMS—thymidylate synthase; dTMP—deoxythymidine monophosphate; DNA—deoxyribonucleic acid; RNA—ribonucleic acid; AICAR—5-aminoimidazole-4-carboxamide ribonucleotide; AICART—5-aminoimidazole-4- carboxamide ribonucleotide transformylase; FAICAR—5-formamidoimidazole-4-carboxamide ribonucleotide; IMP—inosine monophosphate; GMP—guanosine monophosphate; AMPDA—adenosine monophosphate deaminase; ADA—adenosine deaminase; AMP—adenosine monophosphate; CD73—ecto-5′-nucleotidase; A_1_, A_2A_, A_2B_ or A_3_—adenosine receptors. Green arrows represent stimulation, red arrows and sticks represent inhibition, blue arrows represent biochemical conversion, yellow arrows represent migration.

**Table 1 ijms-21-03483-t001:** Methotrexate application as non-oncology drug.

Disease	FDA/EMA Approval	References
Myasthenia gravis	−/−	[17,80,81,82,83,84,85,86,87,88,89,90]
Rheumatoid arthritis	+/+	[14,15,95,100,101,102]
Psoriasis	+/+	[16,68,107,114,115,116]
Crohn’s disease	−/−	[18,117,118]
Sarcoidosis	+/+	[68,114,115,116]
Alzheimer’s disease	−/−	[104,105,106]
Multiple sclerosis	−/−	[19,121,122,123]

**Table 2 ijms-21-03483-t002:** Methotrexate application as anti-tumour agent.

Content of Combination Regimens	Type of Cancer	Reference
LDMTX	childhood ALL	[128]
	PCNSL	[129,130]
	mycosis fungoides	[131,132,133,134]
	HNSCC	[135]
LDMTX + thalidomide	refractory NHL	[136,137]
LDMTX + bleomycin	HNSCC	[138]
LDMTX-LV + PDD + 5-FU		[139]
LDMTX or LDMTX-5-FU		[140]
LDMTX + celecoxib		[141,142]
LDMTX + capecitabine		[143]
LDMTX + interferons	mycosis fungoides	[131,144,145,146,147]
LDMTX + 311 nm UV-B		[148]
LDMTX + radiotherapy		[149]
LDMTX + bleomycin + doxorubicin + topical nitrogen mustard		[150]
LDMTX + CTX + lomustine + etoposide or Adriamycin or vincristine	SCLC	[151]
LDMTX-LV/HDMTX-LV + combination + CTX doxorubicin + vincristine or LDMTX-LV/HDMTX-LV + etoposide + vincristine + hexamethylmelamine		[152]
LDMTX-LV/HDMTX-LV + vincristine + doxorubicin + CTX [MOAC]		[153]
LDMTX + PDD	NSCLC	[154]
LDMTX + mitomycin C + vinblastin + PDD [M-MVP]		[155]
LDMTX-LV / HDMTX-LV	bladder cancer	[10,156,157,158]
LDMTX + PDD		[159]
LDMTX + vinblastine + Adriamycin + cisplatin [MVAC]		[160,161,162,163,164,165]
HDMTX + vinblastine + Adriamycin + PDD [HDMVAC]		
LDMTX + cisplatin + vinblastine [CMV]		
IDMTX	childhood ALL	[166]
HDMTX-LV	childhood ALL	[167,168,169,170]
	PCNSL, widespread NHL	[171,172,173,174,175,176]
HDMTX + citrovorum factor	ALL (B-lineage or T-lineage)	[177]
HDMTX + cytarabine	NHL	[178,179,180]
HDMTX + temozolomide	PCNSL	[181]
HDMTX + rituximab + cytarabine + dexamethasone (R-MAD)	lymphoma	[182]
HDMTX+ bleomycin + Adriamycin + Cytoxan + vincristine		[172]
HDMTX + rituximab +CTX+ doxorubicin + vincristine + prednisolone (R-CHOP)		[183]
HDMTX + rituximab + procarbazine + vincristine (R-MPV)		[184]
HDMTX + rituximab		[185,186,187]
HDMTX-BCG vaccine	HNSCC	[188]
MTX-LV + 5-FU	mycosis fungoides	[189]
MTX-(glutamine)2	brain tumour delivery system	[190]
MTX-(lysine)2		[191]
MTX- magneticNPs-HSA		[192,193,194,195]
MTX- magnetic(iron oxide)NPs		[196]
MTX-polyacrylateNPs		[197]
MTX-chitosanNPs		[198]
MTX-polymersNPs		[199,200]
MTX-protein microspheres		[199,200,201,202,203,204,205,206]
MTX-dendrimers [G5(MTX)n]		[62]
MTX-dendrimers [C225-G5-MTX]		[206,207,208,209]
MTX-dendrimers [PEPE-MTX]		[210]
MTX-PEG		[211]
MTX-EPR		[212]
CTX + MTX + 5-FU [CMF] 5-FU + Adriamycin + CTX [FAC] MTX-LV + tegafur and/or uracil [MUL]	breast cancer (TNBC)	[213,214,215,216,217]
MTX + CTX		[218]
[CMF] + CMFEV + epirubicin or vincristine (rotationally) [CMFEV]		[216]
MTX + mitomycin C [MM]		[219]
MTX + 5-aminoimidazole-4-carboxamide riboside (AICAR)		[220]
MTX + vitamin C		[221]
HDMTX-LV + mechlorethamine + procarbazine	NSCLC	[222]
MTX + PTX + PCNPs [MTX-PMX-PCNPs]		[223]
MTX + pretubulysin	NSCLC (in vitro study)	[224]
MTX + acetylsalicylic acid (ASA)		[225]
MTX-gelatinNPs	pulmonary lung delivery	[226]
HDMTX + 5-FU + vincristine + bleomycin + mitomycin C	blade cancer	[227,228]
HDMTX-LV + doxorubicin + cisplatin + ifosfamide and/or CTX	osteosarcoma	[229]
HDMTX + doxorubicin + PDD [MAP] HDMTX + doxorubicin + PDD + IFN-α-2b		[230]
HDMTX or moderate MTX + Adriamycin + vincristine		[231]
HDMTX + cisplatin + isofosfamide + doxorubicin + muramyl tripeptide encapsulated in liposomes		[232]
HDMTX + bleomycin + CTX + dactinomycin (BCD)		[233]
HDMTX + cisplatin + doxorubicin		[234,235]
HDMTX + trimetrexate; HDMTX + PDD + doxorubicin + ifosfamide + etoposide + zoledronic acid		[236]
MTX + ACT [MACT]	low risk GTN	[237]
HDMTX + etoposide; etoposide + cisplatin /+/ etoposide + MTX + dactinomycin [EP-EMA]	high risk GTN	[238]

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
