# Peer review of "Overview of Dual-Acting Drug Methotrexate in Different Neurological Diseases, Autoimmune Pathologies and Cancers"

_ijms, 2020, doi:10.3390/ijms21103483_

Round 1

Reviewer 1 Report

The authors give a comprehensive overview of current knowledge concerning the structural analogue of  folic acid      i.e., methotrexate, reviewing, in a comprehensive way, its use for the treatment of many types of cancer or serious and resistant forms of autoimmune diseases (and this is a point of strength). A point of weakness is that, in addition to obviously trial studies, FDA guidelines, EMA and original articles, they refer to previous reviews, while in a review these should be taken into consideration only for untreated topics to which the reader should refer. But the real weakness is that this excellent review is completely out of the scope of IJMS, as the molecular mechanisms are treated briefly at the beginning and all the rest is a clinical report.

Author Response

Comments and Suggestions for Authors:

The authors give a comprehensive overview of current knowledge concerning the structural analogue of folic acid i.e. methotrexate, reviewing, in a comprehensive way, its use for treatment of many  kinds of cancer or serious and resistant forms of autoimmune diseases (and this is a point of strength). A point of weakness is that, in addition to obviously trial studies, FDA guidelines, EMA and original articles, they refer to previous reviews, while in a review these should be taken into consideration only for untreated topics to which the reader should refer. But the real weakness is that the this excellent review is completely out of the scope of IJMS, as the molecular mechanisms are treated briefly at the beginning and all the rest is a clinical report.    

Thank you for your comments and suggestions which are valuable and helpful for revising and improving our manuscript.

The part of molecular mechanism of MTX action has been expanded as well as we have added a figure illustrated this mechanism.

Reviewer 2 Report

The paper of Przemysław Koźmiński et al. reviews comprehensively 285 works concerning the application and efficacy of methotrexate in treatment of cancer and in central nervous system disorders. This analysis in detail discusses the mechanism of action and various therapeutics strategies based on this drug and bring a solid portion of facts for scientists and also medical society. I recommend it for publishing after few corrections:

Conclusion: The conclusion is poorly written, and is too general. I don’t know why it is focused mainly on myasthenia gravis. Definitely it should be extensively improved.

It would be beneficial if authors provide the table  similar to Table 1. “Methotrexate application as non-oncology drug”, but for oncological drugs.

It would be also beneficial if authors illustrate the mechanism of action of  methotrexate in the figure (scheme) ( Paragraph 2. Methotrexate – mechanisms of drug action)

Lines 219-233. This paragraph should be moved to the previous part, as it concerns also the anti-inflammatory activity of methotrexate. Otherwise it should be underlined that this issue concerns only high concentrations applied in oncological treatment?

Table 2 should be reorganised because it is very hard to read – especially first column.

There are some linguistic awkwardness in the paper which should be corrected, e.g. DNA strings, inflammation states, circle of great interest, It is also notice worth that, to CO2 and H2O

The shortcuts should be used after first usage of words. Please read the text and correct where appropriate. Also after first use, use only shortcuts.

Author Response

Comments and Suggestions for Authors:

The paper of Przemysław Koźmiński et al. reviews comprehensively 285 works concerning the application and efficacy of methotrexate in treatment of cancer and in central nervous system disorders. This analysis in detail discussed the mechanism of action and various therapeutics strategies based on this drug and bring a solid portion of facts for scientists and also medical society. I recommend it for publishing after few corrections.

Conclusion: The conclusion is poorly written, and is too general. I don’t know it is focused mainly on myasthenia gravis. Definitely it should be extensively improved.

Thank you for your valuable comment. The conclusion has been changed according to the Reviewer's suggestion.

It would be beneficial if authors provide the table similar to Table 1. “Methotrexate application as non-oncology drug”, but for oncological drugs. 

Thank you for the remark. We have modified the table according to the Reviewer’s remark.

It would be also beneficial if authors illustrate the mechanism of action of methotrexate in the figure (scheme). (Paragraph 2. Methotrexate – mechanisms of drug action)

Thank you for the comment. We have added in paragraph 2. and a drawing illustrating the mechanism of drug action.  

Lines 219-233. This paragraph should be moved to the previous part, as it concerns also the anti-inflammatory activity of methotrexate. Otherwise it should be underline that this issue concerns only high concentration applied in oncological treatment?

Thank you for the remark. Indeed, the adverse reaction effects concern mainly the application of high dose methotrexate. We have clearly underlined this problem in the text.

Table 2 should be reorganised because it is very hard to read – especially first column.

Thank you for the suggestion. We've reorganized the table to make it clear and easier to read.

There are some linguistic awkwardness in the paper which should be corrected, e.g. DNA strings, inflammation states, circle of great interest. It is also notice worth that, to CO2 and H20

Thank you for the comment. We have corrected some of the wording in the manuscript, including subscripts.

The shortcuts should be used after first usage of words. Please read the text and correct where appropriate. Also after first use, use only shortcuts.    

Thank you for the remarks. We have read the manuscript carefully and corrected the text according to the Reviewer’s remark.

Reviewer 3 Report

The authors have well elucidated the role and action of mothotrexate in various pathologies. The draft is well conducted, well written and well organized. Anyway I would have suggestions as a minor point:

1-the title should be changed because the space dedicated to the Alzheimer is too little, so the solutions could be :   changing the title by remove "in central nervous system disorders (Myasthenia Gravis, Alzheimer’s disease) and replace  with  in different diseases  neurological and autoimmune pathologies    as cancer.

2- the paragraph from line 78 to 94 should be expanded.
Consequently, the paragraph on inflammatory diseases should be expanded.

3- in line 174 should be a sub title like methotrexate and Alzheimer before the  subtitle methothrexate and other disease.  However the paragraph on alzheimer should be extended.

4- Some more information about multiple sclerosis should be added 

Author Response

Comments and Suggestions for Authors:

 The authors have well elucidated the role and action of methotrexate in various pathologies. The draft is well conduced, well written and well organized. Anyway I would have suggestions as a minor point.

1-the title should be changed because the space dedicated to the Alzheimer is too little, so the solution could be: changing the title by remove ”in central nervous system disorders (Myasthenia Gravis, Alzheimer’s disease) and replace with in different diseases neurological and autoimmune pathologies as cancer.

Thank you for the remark. We have changed the title of the review.

2-the paragraph from 78 to 94 should be expanded. Consequently, the paragraph on inflammatory diseases should be expanded.

Thank you for the important comments. We have expanded the text according to the Reviewer’s remark.

3-in line 174 should be a sub title like methotrexate and Alzheimer before the subtitle methotrexate and other diseases. However the paragraph on Alzheimer should be extended.

Thank you for the suggestion. We have added more information about potentially application of MTX in the treatment of Alzheimer's disease.

4-Some more information about multiple sclerosis should be added.

Thank you for the important comment. We have introduced more information about multiple sclerosis into the paper.

Round 2

Reviewer 1 Report

The authors have implemented information about molecular mechanisms and have prepared an exhaustive figure.